# Exploring the Appeal of Car-Borne Central Control Platforms Based on Driving Experience

Chih-Kuan Lin [1,2], Chien-Hsiung Chen [1] and Kai-Shuan Shen [2,*]

1 Department of Design, National Taiwan University of Science and Technology, No.43, Keelung Rd., Sec.4, Da'an Dist., Taipei City 106335, Taiwan; cklin@mail.fgu.edu.tw (C.-K.L.); cchen@mail.ntust.edu.tw (C.-H.C.)
2 Department of Product and Media Design, Fo Guang University, No.160, Linwei Rd., Jiaosi, Yilan City 262307, Taiwan
* Correspondence: ksshen@gm.fgu.edu.tw; Tel.: +886-39871000 (ext. 25316)

**Abstract:** This study explored drivers' emotion-based impressions of car-borne central control platforms (CBCCPs) for personal-use vehicles. Thus, this preference-based study examined experts' and drivers' opinions regarding the appeal of CBCCPs from the perspective of Miryoku engineering. To this end, this study analyzed data via the EGM (evaluation grid method (EGM) and quantification theory type I. Results: Drivers' preferences for specific CBCCP design characteristics were categorized into the factors "legible", convenient", and "tasteful", which comprised the core of the EGM semantic hierarchical diagram. In addition, the importance of CBCCPs' appeal factors and characteristics was assessed through quantification theory type I. The findings of this study provide valuable insights for designers, manufacturers, and researchers interested in the design of CBCCPs. Additionally, the results of this study can contribute to research on applied psychology, human–computer interactions, and car interface design.

**Keywords:** car-borne user interface; Miryoku engineering; human–computer interaction; psychology; car design

## 1. Introduction

Drivers' interactions with car-borne central control platforms (CBCCPs) have become increasingly complex due to the continuous improvement of cars' functions and performance. Furthermore, the CBCCP plays a critical role in managing an automobile's multiple functions. Hence, a CBCCP that integrates a virtual touchscreen into a physical interface is the best solution for drivers to control a technologically advanced car. Additionally, the CBCCP creates an initial impression on the driver regarding the car's interior, thereby affecting their purchase decision. Hence, it is critical for car manufacturers to design a CBCCP that enables smooth human–computer interaction, conforms to the needs of human ergonomics, and offers an impressive user experience. Hence, the goal of this study was to determine the CBCCP features that are most appealing to drivers, considering the combination of virtual and physical interfaces.

A CBCCP equipped with a touchscreen is currently the most popular solution for technologically advanced cars to provide drivers with control over not only basic functions but also multimedia entertainment systems. However, the virtual touchscreen limits the space of the car's physical interface and affects how the driver operates the CBCCP. Nevertheless, the need for virtual touchscreens is estimated to continue increasing, in tandem with the increase in the market share of electric cars, although CBCCPs' physical interface still plays a critical role for drivers. Shipments of virtual touchscreens are expected to increase by 7.8% and reach 72 million sets by 2023 [1]. This shows that the development of car-borne physical interfaces is gradually decreasing because a touchscreen can integrate various functions into a small display device and thereby change the driver's habits and experiences.

A driver may not have favorable experiences if explicit design norms do not work in concert with the current features on which virtual touchscreens heavily rely [2]. Furthermore, drivers must pay more attention to complex virtual touchscreens while driving. Therefore, advanced virtual touchscreens are not likely to fit all situations and driver needs [3]. Burnett and Porter mentioned that drivers have to rely heavily on their visual focus when operating a touchscreen due to a lack of touching feedback, although touchscreens provide simple and concise instructions [4]. In the past, drivers could carry out simple operations through physical interfaces, such as pressing a button to adjust the sound system's volume. However, currently, drivers need to run complex commands on the touchscreen—e.g., using their eyes to locate where to adjust the sound system's volume and subsequently pressing the correct icon or area. These issues highlight that drivers' experiences in operating central-control platforms are complex and involve several critical fields of research, such as human ergonomics.

Hence, this study explores the design of CBCCPs from several critical perspectives—which are reviewed in Section 2 and can be summarized as follows. This study begins by focusing on drivers' visual and cognitive attention to the car's information system. The study then reviews articles related to human–computer interaction and human ergonomics. Moreover, Miryoku engineering, which is this study's theoretical framework, is also discussed in this section.

Calling for the creation of fully physical interfaces, which would require numerous buttons or knobs in the car's central control platform, is unrealistic because this arrangement could interfere with the driver's decision-making while driving. However, similarly, the trend for fully virtual touchscreens to completely replace physical interfaces should be considered, while complementary measures should be put in place. Hence, hybrid CBCCPs that integrate a virtual touchscreen into a physical interface continue to occupy the mainstream market. Thus, the creation of design norms that make full use of the limited space available in hybrid CBCCPs, aiming to provide drivers with favorable and impressive user experiences, motivates this study. This study explores how a CBCCP can appeal to drivers by using specific design characteristics. It also examines the importance of each design item when developing guidelines for making full use of CBCCPs' limited space. This study's design guidelines can provide valuable insights for manufacturers aiming to provide favorable user experiences for drivers operating CBCCPs.

We argue that the design of an appealing CBCCP should have unique characteristics that align with drivers' preferences through their perception, emotions, and cognition. Hence, we hypothesize that the appeal of CBCCPs can be attributed to specific virtual and physical design characteristics. Hence, in the present study, preference-based Miryoku engineering is used to explore CBCCPs' virtual and physical appeal, based on drivers' emotions. Additionally, the issue of how CBCCPs communicate instantly and accurately with drivers is also investigated in this study. Furthermore, CBCCPs' appeal factors and the characteristics that appeal to drivers are studied, along with the importance of each CBCCP evaluation item. In brief, the above-mentioned three critical issues can be explained through EGM and QTTI from the perspective of Miryoku engineering. Furthermore, this study conducts expert interviews, semantic analysis, questionnaire surveys, and statistical analysis to disclose drivers' preferences through qualitative and quantitative assessments, as determined by the importance of each evaluation item.

The remainder of this paper is organized as follows. The Literature Review section explores the critical topics related to CBCCP design. Third, the Research Objectives section presents the purpose and critical issues considered in this study. Fourth, the Research Methods section depicts the methods used to analyze how consumers' CBCCP preferences can be determined, based on experts' points of view. Fifth, the Discussion section shows the analysis results. Sixth, the Conclusions section reveals the research findings and their implications, as well as this study's limitations and the authors' future research and contributions.

## 2. Literature Review

### 2.1. Drivers' Visual and Cognitive Attention

The issue of drivers' visual and cognitive attention plays an important role in the design of cars' information systems because drivers have to process messages from external conditions and internal information rapidly to make the correct decisions, ensure their safety (and that of any passengers), and reach their destination efficiently. Hence, the design of a car-borne interface that supports drivers in controlling all internal and external conditions is complex and challenging. Thus, several studies have attempted to provide solutions for the myriad obstacles encountered by drivers. Wittmann suggested improving the display screens' location, considering drivers' reaction times and the road conditions [5]. Furthermore, the fact that current display screen configurations are mainly located in the top–middle of CBCCPs confirms Wittmann's study.

### 2.2. Human–Computer Interaction for Drivers

A CBCCP allows the driver to communicate with the vehicle and control it through a computer; as such, it comprises a human–computer interaction (HCI). The Association for Computing Machinery's Special Interest Group on Computer–Human Interaction defined HCI as the discipline cluster that is involved in the design, evaluation, implementation, and use of interactive computing systems [6]. Kantowitz proposed a model that divides human–computer interaction into three elements: the individual, the interface, and the computer [7]. Furthermore, research on HCI explores the communication process among humans, products, computers, and technology. More specifically, the field of HCI comprises messages, responses, and feedback between a machine and an individual. Compared with early mechanical-based cars, vehicles are now highly digitized, thereby increasing the frequency of HCIs. Additionally, the complexity and difficulty of driver–car interactions have also increased. Hence, CBCCP design should focus on improving communication between the driver and the car.

Additionally, the relationship between the dissemination of driving information and drivers' cognition and behavior should be clearly defined so that the driver understands instantly how their vehicle works. Furthermore, the user's interaction with the interface and their environment should be optimized to decrease the gap between the user and the machine and enhance the former's willingness to use CBCCPs. According to the cognitive perspective [8], users' behavior in HCIs can be explained by the way they access pertinent information. Consequently, the success of human–computer interaction depends on the clarity of instructions and feedback, as it determines whether both humans and machines can receive and execute instructions quickly and avoid mistakes.

### 2.3. Ergonomic Psychology Evaluation of the Driver's Experience

In addition to investigating the HCI, examining the ergonomic aspect of the user experience is a critical aspect of this study. Designing CBCCPs requires taking into account their ergonomics to enhance their safety, comfort, productivity, and ease of use for prospective users [9]. Therefore, this study establishes a preference-based engineering system to evaluate whether CBCCPs' current design aligns with drivers' psychological needs.

This study implements both quantitative and qualitative methods, aiming to gain a comprehensive understanding of the CBCCP user experience, although ergonomic evaluations are too subjective to formalize with mathematical algorithms [10]. Therefore, this study conducts a quantitative evaluation based on a qualitative semantic assessment; it examines experts' critical subjective opinions of and statistics on drivers' objective responses. Specifically, the experts' evaluations—which rely heavily on their expertise and personal experience [10]—are examined to establish this study's conceptual framework. Furthermore, semantic evaluation has proven to be an appropriate tool for assessing drivers' subjective impressions, as semantic environment description has been successfully applied elsewhere in cross-cultural comparisons [11]. Thus, in this study, semantic evaluation is applied to gather experts' opinions through in-depth interviews. However, since it is critical to gather

data on numerous drivers, it is necessary to select an appropriate tool to analyze their responses. Although prior studies (e.g., [12]) evaluated drivers' impressions of car interiors by using a simulation procedure, this study uses a questionnaire (developed based on the experts' evaluations) to collect data on drivers' impressions of CBCCPs' design. Thus, in the present study, both qualitative and quantitative methods were used sequentially to integrate experts' opinions with drivers' responses.

*2.4. Miryoku Engineering*

Users' judgments of products depend not only on their rational thoughts but also on their emotions. Hence, several researchers have studied users' emotion-based preferences regarding products. For instance, Khatoon and Rehman declared that consumers' decision-making is strongly affected by the emotions elicited by brand-related stimuli [13]. This underscores the importance of consumers' emotions, which are examined in the present study. The emotion that a product elicits from the user is subjective; hence, product attractiveness plays a key role in users' preferences—which originate from users' systems of judgment. Additionally, users' systems of judgment are formed on the basis of their perceptions and emotions, which comprise the research object of Miryoku engineering. Miryoku (a Japanese term meaning "power of attractiveness" or "appeal") engineering is a technical system of evaluation that integrates psychology, sociology, and aesthetics [14]; it is widely used in the field of design to determine products' attractiveness based on user preferences, as well as to design products in response to such preferences.

Miryoku engineering originated in 1971 when Masato Ujigawa aimed to investigate how to create appealing products, spaces, and technologies. The theory of Miryoku engineering was further developed by Sanui and Inui, who referenced Kelly's personal construct theory to capture users' cognitive concepts vis-à-vis product appeal [15]. Furthermore, in 1991, Massato and Gen held interdisciplinary meetings in the Miryoku Engineering Forum to research problems [16], develop effective theories and methods of product design, and design appealing products [17]. Miryoku engineering is mainly used to examine users' personal preferences; this is usually achieved by comparing two products and interviewing prospective users so that the differences and similarities between said products' characteristics can be determined. Miryoku engineering is currently widely applied in design research [18,19] and is used in the present study to assess the appeal of CBCCPs.

## 3. Research Objectives

Current CBCCPs use a combination of virtual and physical interfaces. Therefore, learning how to integrate the CBCCPs' virtual and psychical interfaces to motivate drivers' intentions to use them could be valuable for both researchers and manufacturers. Furthermore, drivers' judgment systems—influenced by complex emotional and perceptual systems—are examined to identify their preferences regarding CBCCPs. Thus, in the present study, drivers' emotions in response to CBCCP design characteristics are investigated, while the importance of critical design factors and characteristics is determined to assess CBCCPs' appeal.

## 4. Research Methods

In the present study, Miryoku engineering is applied to evaluate user preferences regarding CBCCPs and serves as this study's theoretical framework. This study collected experts' opinions regarding CBCCP design characteristics, which were translated into intuitive concepts so that general drivers' responses could be evaluated and analyzed based on the experts' subjective points of view. Hence, the relationship between drivers' impressions and CBCCP design can be determined based on Miryoku engineering. Subsequently, the general drivers' needs and preferences were determined and were then used to plan the design of CBCCPs. Specifically, this study uses the EGM and QTTI to implement Miryoku engineering and suggest improvements to CBCCPs to make them more appealing.

This study examines CBCCPs' appeal based on Miryoku engineering, which can be practiced by recognizing semantic differences and conducting multivariate analysis [20]. First, the EGM is used to acquire experts' opinions and establish an evaluation structure, while content analysis is performed to develop a universal guideline for CBCCP design. Furthermore, this study conducts a questionnaire survey based on these design guidelines regarding EGM to investigate drivers' preferences. Finally, the questionnaire survey is statistically analyzed via QTTI to determine the importance of each appeal factor and design characteristic.

*4.1. Evaluating Experts' Opinions Using the EGM*

This study designed the expert interviews following the EGM and the repertory grid method [21]. In the beginning, participants were asked to compare the designs of various CBCCPs and to select which ones they preferred or disliked. Subsequently, the questions clarified the participants' reasons for their preference for specific CBCCPs. Furthermore, this study collected semantic data from article reviews and expert interviews, and then codified them by distinguishing between abstract expressions and specific lexicons. This method conformed to the EGM, which has been widely applied to organizing participants' answers in design research [22,23]. The results of the EGM are summarized in a hierarchical diagram.

The EGM was used to identify the factors and characteristics that make CBCCPs appealing to users, according to experts' opinions. The EGM applies a semantic analysis technique to extract experts' opinions expressed in interviews or articles. Subsequently, the relationship between CBCCPs' appeal factors and design characteristics can then be constructed and presented through a hierarchical diagram. Hence, this study collected experts' opinions using two sources: articles and in-depth interviews.

At the beginning of the EGM procedure, this study gathered data on descriptive phrases and specialized nouns in authoritative articles published in auto magazines, professional web sites, and columns that reviewed CBCCPs' design. This revealed the factors and characteristics that make CBCCPs appealing to users. Then, the initial knowledge base of CBCCPs' design was mapped out; this framework served as the foundation of the rest of the investigation and expert interviews.

This study designed the expert interviews following the EGM and the repertory grid method [21]. In the beginning, participants were asked to compare the designs of various CBCCPs and to select which ones they preferred or disliked. Subsequently, the questions clarified participants' reasons for their preference for specific CBCCPs. Furthermore, this study collected semantic data from article reviews and expert interviews and codified them by distinguishing between abstract expressions and specific lexicons. This method conformed to the EGM, which has been widely applied to organize participants' answers in design research [16,22]. The results of the EGM are summarized in a hierarchical diagram.

In-depth interviews offer a more appropriate way to tap into experts' professional knowledge, compared with multiple-choice surveys. However, a systematic and reliable way to extract experts' knowledge from interviews is necessary. Therefore, this study used the EGM to analyze experts' opinions regarding CBCCPs' design. The results of the EGM (which were used to determine critical items of CBCCP design) are presented through interpretive structural modeling. This structural model shows the captured appeal factors and characteristics in a hierarchical arrangement. The EGM-based procedure was conducted as follows.

First, the authors hired three research assistants and trained them to distinguish between technical descriptions (e.g., "full touchscreen interface") and abstract emotive phrases (e.g., "clear"). Hence, the three research assistants generalized the descriptive phrases collected from the 10 interviews concerning CBCCP design, based on the EGM.

Second, this study individually interviewed five male and five female experts between the ages of 26 and 65, all of whom had at least five years of driving experience and who were familiar with vehicle sales or design (Table 1). In addition, 58 sample cards with pictures presenting CBCCPs' critical design characteristics were prepared for the

interviews. In individual one-hour interviews, the experts were asked to express their preferences regarding CBCCP design. Furthermore, a grouping method was used, in which the interviewees classified the 58 sample cards into three stacks so that their preferences could be more easily recognized. Subsequently, these experts explained why they grouped these sample cards.

**Table 1.** The detailed description of the interviewees for EGM evaluation.

| Sex | Ages | Working Years | Position | Responsible for Business |
|------|------|------|------|------|
| Male | 65 | 45 | General Manager | Vehicle sales, including electric cars |
| Male | 62 | 42 | Senior Manager | Vehicle sales |
| Male | 54 | 35 | Manager | Department of electric car design |
| Male | 50 | 28 | Designer | Department of vehicle design |
| Male | 45 | 21 | Editor | Car reviews and analysis |
| Female | 48 | 21 | Editor | Car reviews and analysis |
| Female | 45 | 20 | Editor | Car reviews and analysis |
| Female | 42 | 20 | Designer | Department of vehicle design |
| Female | 40 | 18 | Designer | Department of electric car design |
| Female | 26 | 15 | Salesperson | Vehicle sales |

Third, this study identified 76 emotional words and 131 technical phrases that were used to evaluate the design of CBCCPs according to the preferences of 10 experts. Then, the research assistants processed the identified words and phrases using a judging mechanism to distinguish whether they were "upper-" or "lower-level" concepts.

Fourth, EGM-based content analysis was used to capture the experts' preferences, as expressed during the interviews. Furthermore, the technical phrases gathered from the interviews were classified as either upper- or lower-level concepts. The upper- and lower-level concepts were used to construct a hierarchical structure of semantics to present the results of the EGM. The upper-level concepts indicated the words that were used to convey consumers' abstract feelings, which usually corresponded to adjectives. Conversely, the lower-level concepts comprised phrases that indicated specific design characteristics, which corresponded to technical nouns.

Fifth, the original evaluation items could be developed through the evolution of upper- and lower-level concepts. More specifically, the original evaluation items could be presented by the continuous convergence of similar "upper-level" and "lower-level" concepts, so that every evaluation item is unique.

After the above procedures were completed, a hierarchical diagram was drawn up to show the results of the EGM. Hence, the CBCCPs' appeal factors and specific marketing characteristics were determined.

### 4.2. Evaluating Consumers' Responses Using QTTI

After the EGM was applied to determine experts' general preferences, this study used QTTI to measure the importance of each evaluation item to assess the consumers' emotional tendencies. Furthermore, the upper- and lower-level items were evaluated via QTTI. QTTI can be used to predict the relationship between a response value and categorical values [23]. Hence, the weight of each item affecting the users' preferences can be calculated using QTTI [23–25]. In brief, Miryoku engineering can measure consumers' emotional preferences for a given product's features and characteristics.

Furthermore, this study developed a questionnaire using the hierarchical structure of the EGM. The questionnaire was designed based on the results of the EGM; the evaluation items that appeared most often were selected for inclusion in the consumer survey. Specifically, this questionnaire was based on a three-level hierarchy that included the upper- and lower-level terms, as well as the original evaluation items. The questions were designed to be as concise as possible. Table 2 shows the development of the questionnaires.

**Table 2.** The setting of the level-based construction of the questionnaire.

| Level of Questionnaire | First Level | Second Level | Third Level |
|---|---|---|---|
| Type of question | Original evaluation item | Upper level (an image) | Lower level (a specific trait) |
| The example of a question subject | Convenient | Intuitive | Integrated information system |

Drivers' emotional tendencies were assessed through a questionnaire survey. This study purposively sampled a group of consumers familiar with the design of CBCCPs. The respondents were asked to invite qualified individuals to participate in the survey in order to establish a larger sample and ensure the reliability of the questionnaire data.

This study investigated consumer preferences by distributing 385 questionnaires to the target subjects, including experienced users and owners of CBCCPs. Subsequently, 355 questionnaires were returned, of which 43 questionnaires were rejected. Thus, 312 (81%) valid questionnaires were included in the analysis. The respondents comprised 162 men and 150 women (aged 22–56 years). The questionnaire data were analyzed using QTTI.

## 5. Analysis and Results

### 5.1. Evaluating Experts' Preferences Using the EGM

This study applied the EGM to derive the general design guidelines from the experts' opinions. The results of the analysis are presented using a hierarchical diagram that shows the structure of the evaluation items for CBCCPs. The EGM enables a distinction to be made between abstract impressions and specific design characteristics, as well as their classification into a three-level hierarchy (comprising the original evaluation items, as well as upper- and lower-level concepts). The hierarchical diagram shows the results of the EGM and illustrates which CBCCP characteristics are appealing to consumers. Figure 1 shows how the evaluation items were identified during the interviews. Figure 2 shows the hierarchical structure that was developed, based on the interviews. Figure 2 also lists all the CBCCP appeal factors and characteristics, as well as the relationships among them. Thus, this study determines which CBCCP characteristics appeal to consumers. In the present study, the design characteristic "rectangular 10–12-inch display screen" was found to be the optimal size for the display screen to engage the drivers' visual attention, according to the experts' recommendation. Additionally, the design characteristic "indicator light" engages drivers' cognition to identify instant messages. These design characteristics play a role in how drivers recognize the information they are given regarding the car's internal status and the road's external conditions.

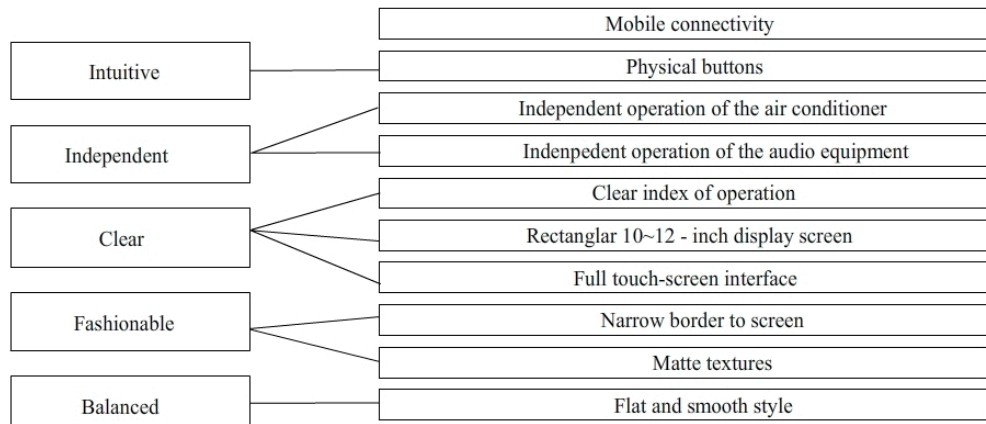

**Figure 1.** An example of one participant's evaluation structure. Note: The No. 5 evaluation structure was constructed by retrieving the answers from an experienced user who is 48 years old.

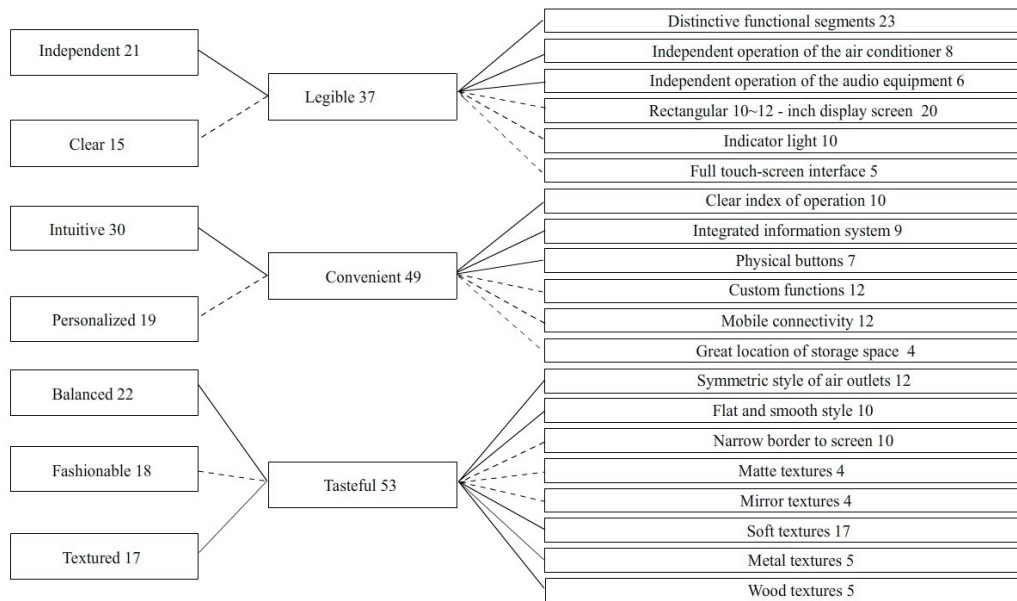

**Figure 2.** The hierarchical diagram of the total set of experts' preferences for car-borne central control platforms, as determined by EGM. Note: The solid lines and dashed lines are used to help readers distinguish different evaluation items.

The questionnaire was constructed based on the hierarchical diagram (Figure 2) and comprised three sections, grouped according to the original evaluation items, including "legible", "convenient", and "tasteful". Furthermore, each section consisted of two types of questions asking consumers about upper- and lower-level concepts. The upper-level concepts comprised "balanced", "fashionable", "textured", "intuitive", "personalized", "independent", and "clear". Conversely, the lower-level concepts included "symmetric style of the air outlets", "flat and smooth style", "narrow border to screen", "matte textures", "mirror textures", "soft textures", "metal textures", "wood textures", "clear index of operation", "integrated information system", "physical buttons", "custom functions", "mobile connectivity", "great location of storage space", "distinctive functional segments", "independent operation of the air conditioner", "independent operation of the audio equipment", "rectangular 10–12-inch display screen", "indicator light", and "full touchscreen". All evaluation items that were ranked according to the EGM (Table 3) were used to investigate consumers' preferences for CBCCPs through the questionnaire.

**Table 3.** Original impressions of the top three evaluation items, selected according to the number description in the EGM hierarchical chart and their reasons. Note: The numbers out of the brackets indicate the number of times that the same opinion appeared.

| Original Images | Upper Level | Lower Level |
|---|---|---|
| Tasteful 53 (1) | Balanced 22 (2) | Symmetric style of air outlets 12 (4)<br>Flat and smooth style 10 (5)<br>Narrow border to screen 10 (5) |
| | Fashionable 18 (5) | Matte textures 4 (11)<br>Mirror textures 4 (11) |
| | Textured 17 (6) | Soft textures 17 (3)<br>Metal textures 5 (10)<br>Wood textures 5 (10) |
| Convenient 49 (2) | Intuitive 30 (1) | Clear index of operation 10 (5)<br>Integrated information system 9 (6)<br>Physical buttons 7 (8) |
| | Personalized 19 (4) | Custom functions 12 (4)<br>Mobile connectivity 12 (4)<br>Great location of storage space 4 (11) |

**Table 3.** *Cont.*

| Original Images | Upper Level | Lower Level |
|---|---|---|
| Legible 37 (3) | Independent 21 (3) | Distinctive functional segments 23 (1)<br>Independent operation of the air conditioner 8 (7)<br>Independent operation of the audio equipment 6 (9)<br>Rectangular 10~12-inch display screen 20 (2) |
| | Clear 15 (7) | Indicator light 10 (5)<br>Full touchscreen 5 (10) |

Note: number in the "( )" indicates ranking.

### 5.2. QTTI Analysis for Surveying Consumers' Preferences

This subsection shows the results of the quantitative analysis, which determined the importance of each factor to the design of a CBCCP. Data from the questionnaire survey were collected and analyzed using QTTI. Quantitative analysis was performed using the following procedure. First, the original evaluation items of the EGM were judged to be reliable, based on their coefficients of determination. Second, the upper-level items of the EGM were weighted to determine the relationship strength, based on the numerical values of the partial correlation coefficients. Third, the importance of lower-level items was determined through category scores. Fourth, this study analyzed QTTI by transferring the mathematical formula to Excel Macro.

The results of the quantitative analysis based on QTTI are presented in Tables 3–5. Additionally, the appeal factors of "legible", "convenient", and "tasteful", as well as their corresponding evaluation items, were checked using the partial correlation coefficients, the category scores, and the coefficient of determination.

The factor "legible" proved to be highly important for consumers, as its coefficient of determination ($R^2$ = 0.528) showed high reliability according to QTTI (Table 4). Moreover, CBCCPs were perceived as "legible" by consumers, based on the characteristics "independent" and "clear". Furthermore, the upper-level concepts of "independent" and "clear" had a stronger influence on CBCCPs being perceived as "legible", based on their partial correlation coefficients (0.558 and 0.667, respectively). The factor "independent operation of the audio equipment" had the highest scores among all categories; thus, it had a strong positive impact on the perception of the concept "independent".

**Table 4.** The partial correlation coefficients, the category scores, and the coefficient of determination for the factor of "legible".

| Items | Categories | Category Scores | Partial Correlation Coefficients |
|---|---|---|---|
| Independent | | | 0.558 |
| | Distinctive functional segments | −0.024 | |
| | Independent operation of the air conditioner | 0.124 | |
| | Independent operation of the audio equipment | 0.162 | |
| Clear | | | 0.667 |
| | Rectangular 10~12-inch display screen | −0.137 | |
| | Indicator light | 0.090 | |
| | Full touch-screen interface | 0.033 | |
| C = 0.828<br>R = 0.722<br>$R^2$ = 0.528 | | | |

**Table 5.** The partial correlation coefficients, the category scores, and the coefficient of determination for the factor of "handy".

| Items | Categories | Category Scores | Partial Correlation Coefficients |
|---|---|---|---|
| Intuitive | | | 0.704 |
| | Clear index of operation system | −0.035 | |
| | Integrated information system | −0.052 | |
| | Physical buttons | 0.116 | |
| Personalized | | | 0.711 |
| | Custom functions | 0.120 | |
| | Mobile connectivity | −0.034 | |
| | Great location of storage space | −0.057 | |

C = 0.900
R = 0.784
$R^2$ = 0.615

The factor "convenient" was highly important for consumers, as its coefficient of determination ($R^2$ = 0.615) showed high reliability according to QTTI (Table 5). This shows that CBCCPs were perceived as "convenient" by consumers, based on the reasons "intuitive" and "personalized". Moreover, "intuitive" and "personalized" had a strong influence on CBCCPs being perceived as "convenient", based on their partial correlation coefficients (0.704 and 0.711, respectively). Moreover, the "customized function" exhibited the highest scores among all categories; thus, it had the most positive influence on the evaluation item "personalized".

The factor "tasteful" was highly important to consumers, as its coefficient of determination ($R^2$ = 0.531) showed high reliability according to QTTI (Table 6). Additionally, CBCCPs were perceived as "tasteful" based on the characteristics "balanced", "fashionable", and "textured". Furthermore, the upper-level concepts "fashionable" and "textured" had a strong influence on CBCCPs' perception as "convenient", based on their partial correlation coefficients (0.620 and 0.536, respectively). "Matte texture" exhibited the highest scores among all categories, which indicates that it had the most positive influence on the CBCCPs being perceived as "fashionable" by consumers.

**Table 6.** The partial correlation coefficients, the category scores, and the coefficient of determination for the factor of "tasteful".

| Items | Categories | Category Scores | Partial Correlation Coefficients |
|---|---|---|---|
| Balanced | | | 0.367 |
| | Symmetric style of air outlets | 0.038 | |
| | Flat and smooth style | −0.004 | |
| Fashionable | | | 0.620 |
| | Narrow border to screen | 0.035 | |
| | Matte textures | 0.055 | |
| | Mirror textures | −0.014 | |
| Textured | | | 0.536 |
| | Soft textures | −0.002 | |
| | Metal textures | 0.048 | |
| | Wood textures | −0.112 | |

C = 0.804
R = 0.728
$R^2$ = 0.531

## 6. Discussion

Figure 2 depicts the results of the EGM through a hierarchical diagram and illustrates the CBCCPs' appeal factors, including "legible", "convenient", and "tasteful". Additionally, the customers' reasons underlying their preferences, as well as the characteristics that correspond to the appeal factors, are listed in the hierarchical diagram. Subsequently, the evaluation items that were selected based on their frequency in the EGM were identified based on the consumers' reactions. The questionnaire data were assessed using QTTI to assess each evaluation item's importance.

This study proved that all three appeal factors play a critical role in the design of CBCCPs, as evidenced by their reliability scores: "legible" ($R^2 = 0.528$), "convenient" ($R^2 = 0.615$), and "tasteful" ($R^2 = 0.531$). The finding that "personalized" and "intuitive" have higher partial correlation coefficients (0.711 and 0.704, respectively) than other items indicates that these two items not only play a critical role in the factor "convenient" but also have a strong impact on consumers' impressions of CBCCPs. Furthermore, the impression "personalized" reflects the consumers' preference for CBCCPs that align with their habits and behaviors so they feel comfortable while driving their car. Accordingly, "customized functions" is the most critical design characteristic for strengthening customers' impressions of "personalized" CBCCPs. Hence, integrating numerous functions into a CBCCP to cater to the consumers' need for a personalized and convenient interface is essential for car manufacturers. Additionally, "intuitive" indicates the need for CBCCPs to be used easily in both ordinary and urgent conditions. Notably, the category "physical buttons" exhibited higher scores (0.116), compared with other categories associated with the concept "intuitive". Before these results came out, we doubted whether drivers still needed some antiquated types of interfaces, such as "physical buttons". However, this result highlights that traditional physical interfaces still play a critical role in the design of CBCCPs, as they provide drivers with intuitive feedback.

Both "independent operation of the audio equipment" (0.162) and "independent operation of the air conditioner" (0.124) exhibited high scores. This indicates that drivers prefer the interfaces of certain functions to operate independently so they feel "legible". Consequently, car manufacturers could benefit from placing the functions of the air conditioner and audio equipment in independent interfaces to improve consumers' "legible" experience. Therefore, taking into account drivers' impressions of CBCCPs as "legible", "convenient", "independent", "intuitive", and "personalized", as well as the design characteristics of "customized functions", "physical buttons", "independent operation of the audio equipment", and "independent operation of the air conditioner" could greatly improve consumers' experience regarding safety and ease-of-use [9].

Besides "legible" and "convenient", "tasteful" ($R^2 = 0.531$) is an ignorable impression that is used to attract consumers so they can feel that the design of CBCCPs is "balanced," "fashionable", and "textured". Furthermore, drivers' preference for "tasteful" and "fashionable" interfaces indicates that CBCCPs should be aesthetically pleasing and novel for the consumer; this finding is in line with prior studies, which have found aesthetics to be a critical factor in consumers' acceptance intentions [26].

In brief, according to this study's hierarchical model (Figure 2) and findings (Tables 3–5), car manufacturers should include critical CBCCP characteristics (Table 7) such as "physical buttons" and "customized functions" to improve consumers' impressions of the interface as being "tasteful", "convenient", and "legible". In contrast, it is interesting that physical buttons seem no different from haptic touchscreens (buttons) for evaluating 23 truck drivers' acceptance and satisfaction of hardware operations in vehicle cockpits, according to the study by Schölkopf et al.; however, these results were probably caused by the practical necessities of working. Compared to truck drivers, general CBCCP users prefer "physical buttons" because of their established habits of use, which could influence their impression of "intuitive." This comparison reveals that different purposes of use could lead to various results in a study evaluating human–computer interaction [27].

**Table 7.** All positive evaluation factors, items, and categories.

| Appeal Factors | Items | Categories |
|---|---|---|
| Legible | Independent | Independent operation of the air conditioner |
| | | Independent operation of the audio equipment |
| | Clear | Indicator light |
| | | Full touch-screen interface |
| Handy | Intuitive | Physical buttons |
| | Personalized | Custom functions |
| Tasteful | Balanced | Symmetric style of air outlets |
| | Fashionable | Narrow border to screen |
| | Textured | Matte textures |
| | | Metal textures |

## 7. Conclusions

### 7.1. Implications

This study explored CBCCP design from the perspective of Miryoku engineering and identified the appeal factors and characteristics that influence consumers' preferences for CBCCPs. As virtual interfaces are now ubiquitous in vehicles for the control of their complex systems, physical interfaces are gradually being phased out, as they occupy a large amount of space in the central control platform and control limited functions for driving. However, some physical elements, such as buttons, are still included in CBCCPs. Thus, physical elements still play a critical role in the design of CBCCPs. The integration of limited physical elements into virtual CBCCPs is a critical issue for car manufacturers.

The results of this study indicate that drivers' perceptions of the CBCCP as "tasteful", "convenient", and "legible" originate from design characteristics that combine the virtual and physical elements of CBCCPs; hence, hybrid CBCCPs could provide consumers with a unique user experience and, thus, seem more appealing. Additionally, this study indicates that ensuring that CBCCPs are intuitive and easy to use, while providing clear instructions, could increase consumers' willingness to engage with them. The present study's findings also show general consumer responses to a questionnaire designed and based on the experts' point of view—which originates from the latter's user experience. Furthermore, these results indicate that the evaluation of CBCCPs' ergonomic qualities depends on the experts' personal experience [10].

The results of this study can help readers to understand CBCCPs' appeal. Car dealers and manufacturers attempt to attract consumers by designing appealing exterior and interior features, to increase the latter's willingness to purchase. Furthermore, a car's interior features could be crucial in a consumer's purchase decision, while the central control platform is a central design element. CBCCP design involves several research fields, including ergonomics and aesthetics. For instance, the drivers' impressions of CBCCPs as "convenient" and "legible" are related to ergonomics and reflect the consumers' need for an interface that is easy to use in urgent situations. The drivers' impressions of CBCCPs as "tasteful" indicate the consumers' need for aesthetically pleasing features. Moreover, the appropriate design of a vehicle's CBCCP is not only beneficial to driver safety but also improves the user experience.

### 7.2. Contributions

The results of this study contribute to the fields of ergonomics and design. Undoubtedly, the development of electric cars is an important trend in automobile design. Working in tandem, digitized interfaces are becoming increasingly common in CBCCPs. Hence, the integration of physical features into digitized central control platforms in electric cars could become the focus of our future research.

*7.3. Limitations and Future Works*

This study had some limitations. First, this study analyzed Taiwanese experts' and consumers' opinions regarding CBCCP design. Therefore, the results of this study may not be generalizable to other regions. Second, this study focused on personal-use vehicles and excluded those vehicles intended for special purposes. Hence, our future work plans are to investigate more CBCCP users using this research structure in some representative areas, such as in North America and Europe, to make the results of the study more reliable. In addition, we will focus on the interior design of an electric vehicle that integrates a large number of new technological and innovative elements.

**Author Contributions:** Conceptualization, C.-K.L. and K.-S.S.; methodology, K.-S.S.; software, K.-S.S.; validation, C.-H.C. and K.-S.S.; formal analysis, K.-S.S.; investigation, K.-S.S.; resources, C.-K.L.; data curation, C.-H.C.; writing—original draft preparation, K.-S.S.; writing—review and editing, C.-H.C.; visualization, C.-K.L.; supervision, C.-H.C.; project administration, C.-K.L. All authors have read and agreed to the published version of the manuscript.

**Funding:** This research received no external funding.

**Institutional Review Board Statement:** The study was conducted in accordance with the Declaration of Helsinki and was approved by the Institutional Review Board (or Ethics Committee) of NAME OF INSTITUTE.

**Informed Consent Statement:** Informed consent was obtained from all subjects involved in the study.

**Data Availability Statement:** The data that support the findings of this study are available from the corresponding author upon reasonable request.

**Acknowledgments:** The researchers would like to thank those who contributed to the completion of this research. In addition, we would like to thank all the participants for helping us conduct interviews and answer questionnaires to make this study more complete.

**Conflicts of Interest:** The authors declare no conflict of interest.

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
