# Peer review of "Exploring the Appeal of Car-Borne Central Control Platforms Based on Driving Experience"

_mti, doi:10.3390/mti7110101_

Round 1

Reviewer 1 Report

Comments and Suggestions for Authors

In this paper, the authors propose a review-like study to analyze the effectiveness of car-bone central control platforms for drivers. The study involves different sub-topics, e.g., multimodal interaction, psychology, and Miryoky engineering methodology.

The overall quality of the manuscript seems quite high. Thus, only some suggestions are provided for improving the readability.

 There are some repetitions in the manuscript. E.g. the first time EGM and QTTI are introduced is in the first section, but they are explained in section 4. Thus, we suggest the authors restructure some parts to avoid repetitions.

Some figures could support the reader in better understanding the environments, the interactions, and the overall analyzed topic.

Tables 3, 4, and 5 could be more clear. Please restyle them.

Two novel tables should be introduced:

1) The population involved (details for each participant or group of participants)

2) Collected results in a unique structure (combine tables 3, 4, and 5)

These will support the reader to focus on some numerical details collected in this research.

From line 472, the limitations are introduced. Thus, we expect that some future improvements could be introduced.

Author Response

Reviewer 1’ suggestions Authors’ responses
The overall quality of the manuscript seems quite high. Thus, only some suggestions are provided for improving the readability. Thanks for reviewer 1’s appreciation to motivate our work. We revised our article according to your suggestion.
There are some repetitions in the manuscript. E.g. the first time EGM and QTTI are introduced is in the first section, but they are explained in section 4. Thus, we suggest the authors restructure some parts to avoid repetitions. Thanks for reviewer 1’s suggestion. I removed the descriptions of EGM and QTTI from the first section to section 4 and integrated them to this article.
Some figures could support the reader in better understanding the environments, the interactions, and the overall analyzed topic. Tables 3, 4, and 5 could be clearer. Please restyle them. Thanks for reviewer 1’s suggestion. I restyle Tables 3, 4, and 5 by adding a row for each evaluation item to make these tables more readable. 

Two novel tables should be introduced:

1) The population involved (details for each participant or group of participants)

Thanks for reviewer 1’s suggestion. We created Table 2 to present the detailed description of the interviewees for EGM evaluation.

2) Collected results in a unique structure (combine tables 3, 4, and 5)

Thanks for reviewer 1’s suggestion. We created Table 7 to show all the positive evaluation factors, items, and categories

From line 472, the limitations are introduced. Thus, we expect that some future improvements could be introduced. Thanks for reviewer 1’s suggestion. We created section 7.3 to describe the limitations and future works

Reviewer 2 Report

Comments and Suggestions for Authors

The article analyzes the drivers' emotion-based impressions of car-borne central control platforms (CBCCPs) personal-use vehicles. It covers an interesting topic with practical implications relevant to the road safety industry. In the following, I will make some recommendations that I consider can improve the quality of the paper.

The introduction and literature review are quite complete and adequately contextualize the issue for the reader.

In addition, the methods and results are clear and adequately presented. However, I recommend further development of the discussion section, as this should not only be a summary of the results but should also contrast the data obtained with other research. Thus, I recommend that this section be developed by answering the questions: were the results in accordance with expectations, are the results congruent with other similar research? And, if not, what elements explain the discrepancies that have occurred?

A specific subsection should also be included with the limitations of the study and future lines of research.

Author Response

Reviewer 2’ suggestions Authors’ responses
The article analyzes the drivers' emotion-based impressions of car-borne central control platforms (CBCCPs) personal-use vehicles. It covers an interesting topic with practical implications relevant to the road safety industry. In the following, I will make some recommendations that I consider can improve the quality of the paper. Thanks for reviewer 2’s appreciation to motivate our work. We revised our article according to your suggestion.
The introduction and literature review are quite complete and adequately contextualize the issue for the reader. Thanks very much for reviewer 2’s appreciation.
In addition, the methods and results are clear and adequately presented. However, I recommend further development of the discussion section, as this should not only be a summary of the results but should also contrast the data obtained with other research. Thus, I recommend that this section be developed by answering the questions: were the results in accordance with expectations, are the results congruent with other similar research? And, if not, what elements explain the discrepancies that have occurred? Thanks for reviewer 2’s appreciation. We compared our results with Schölkopf et al.’s study to explain how the differences between them in the end of section 6. In addition, we expressed our reactions for the result that is not congruent with our original assumption in the end of the second paragraph in section 6. 
A specific subsection should also be included with the limitations of the study and future lines of research. Thanks for reviewer 2’s appreciation. We created subsection 7.3 to present the limitations of this study and our future works
